# Prevalence and molecular characterization of human bocavirus-1 in children and adults with influenza-like illness from Kunming, Southwest China

Yanhong Sun,[1] Lili Jiang,[1] Yaoyao Chen,[1] Zhaosheng Liu,[1] Meiling Zhang,[1] Xiaonan Zhao,[1] Xiaoyu Han,[1] Lifen Zhang,[1] Xiaoqing Fu,[1] Jienan Zhou[1]

**ABSTRACT**  Human bocavirus-1 (HBoV-1) has been associated with respiratory infections in both children and adults, often presenting symptoms similar to those of influenza. Understanding the prevalence and molecular characteristics of HBoV-1 in individuals with influenza-like illness (ILI) is essential for enhancing the diagnosis and treatment of respiratory infections in Kunming, Southwest China. Between December 2017 and December 2023, demographic and clinical data, along with respiratory tract specimens from individuals aged 0 to 97 years with ILI, were collected at three sentinel hospitals in Kunming. Each specimen was tested for 18 respiratory viruses, and the positive rates of HBoV-1 across different age groups were analyzed. Amplification of the near-complete HBoV genome was achieved through three overlapping fragments, followed by next-generation sequencing (NGS) and phylogenetic analysis. A total of 20,181 respiratory samples were collected from patients aged 1 month to 97 years presenting with ILI symptoms between December 2017 and December 2023, with HBoV detected in 0.8% of the samples. The prevalence was 1.0% (165/16,406) in children and 0.1% (3/3,775) in adults, with a significantly higher detection rate in pediatric patients (<18 years old) compared to adults ($\geq$18 years old) ($P < 0.001$). Among the 168 HBoV-positive participants, 165 (98.2%) were children under 18 years, while 3 (1.8%) were adults. Genome-wide phylogenetic analyses indicated that HBoV-1 was the predominant genotype, showing that the HBoV-1 strains circulating in Kunming are closely related to strains from other regions of China and globally. Our findings confirm the prevalence of HBoV-1 in individuals with ILI in Kunming and provide valuable insights into the molecular characteristics of HBoV-1 in this region. Further studies are necessary to explore the clinical implications of HBoV-1 infection and its role in respiratory illnesses.

**IMPORTANCE**  Viral respiratory infections are a leading cause of morbidity- and mortality-associated influenza-like illness (ILI) cases. It is estimated that there are several billion cases of ILI globally each year. Monitoring data from China in 2023 indicate that there are approximately 17 million cases of ILI nationwide. In the United States, the annual incidence of ILI ranges from 9 to 49 million cases. Human bocavirus-1 (HBoV-1) has been identified as a causative agent of ILI. The global prevalence of HBoV-1 respiratory infections varies from 1% to 56.8%, with the majority of studies focusing on pediatric populations; however, research including a broader age range is limited. Currently, the prevalence of HBoV-1 in the Kunming area is not well characterized, and its molecular features remain inadequately described. This study aims to analyze the prevalence of HBoV-1 among ILI cases in Kunming, encompassing both pediatric and adult patients. We present 107 complete genomic sequences of HBoV-1 strains obtained from three ILI sentinel hospitals in the region. Furthermore, we conducted phylogenetic analysis, homology comparisons, and assessments of nucleotide and

**Peer Reviewers** Ali Hafedh Abbas, University of Baghdad, Baghdad, Iraq; Nikhil Chakravarty, University of California Los Angeles, Porter Ranch, California, USA

Address correspondence to Jienan Zhou, 1191087570@qq.com.

The authors declare no conflict of interest.

amino acid substitution site variations. These findings provide important insights for further investigations into HBoV-1 and its epidemiological significance.

**KEYWORDS**    HBoV-1, prevalence, molecular characterization, children and adults, ILI

Respiratory viral infections are among the most prevalent infectious diseases worldwide and represent a significant public health concern (1). They are causal agents of influenza-like illness (ILI) and acute respiratory infections in children and adults, leading to considerable morbidity and mortality year after year (2). It is estimated that approximately 75%–90% of ILI cases globally are caused by viruses. Influenza viruses and respiratory syncytial viruses are the most commonly detected viral pathogens. Recently, human rhinovirus and human bocavirus-1 (HBoV-1), identified through molecular methods, are widely prevalent and geographically distributed, frequently associated with severe respiratory disease following primary infections, particularly in young children (1).

HBoVs are classified within the proposed genus *Bocaparvovirus* in the family *Parvoviridae*. These small, non-enveloped, icosahedral viruses that can be positive- or negative-sensed are approximately 25 nm in diameter with a 5.3 kb single-stranded DNA genome containing three open reading frames (ORFs). The first two sequential ORFs encode nonstructural proteins NS1 and NP1, while the third downstream ORF encodes three viral capsid proteins: VP1, VP2, and VP3, with VP3 being the predominant capsid protein (3, 4). These three VP proteins are located within the same coding region and differ in the positioning of their start codons, with VP2 utilizing a non-canonical start codon (4). Virus replication relies on the host cell machinery. The Bocavirus genome features two terminal hairpin structures at both ends of its single-stranded DNA, which are essential for viral replication (5). No virus was isolated or cultivated; only fragmented viral sequences have been reported (6–8). Four genotypes of HBoV have been identified based on the genetic diversity of the VP1 gene. 19 years ago, HBoV-1 was identified by Allander et al. (9) utilizing a novel molecular assay for screening respiratory specimens collected from Swiss pediatric participants suffering from respiratory tract infection (RTI) of unknown etiology. Since the discovery of the virus, it has been observed that HBoV-1 is the fourth most common virus in respiratory samples (10). Infections with HBoV-1 are characterized by influenza-like symptoms and lower respiratory tract manifestations (11, 12). Although HBoV-1 is most commonly detected in the respiratory pathway and occasionally in stool, it can also be found in blood, cerebrospinal fluid, and tonsillar tissues. By 2010, three additional genotypes—HBoV2, HBoV3, and HBoV4—were discovered in human fecal samples from children with gastrointestinal complaints (13), though these have been rarely tested in respiratory specimens. Some data suggest a link between HBoV2 and gastroenteritis, but HBoV3 and HBoV4 have not been conclusively associated with any specific clinical illness.

Since their discovery in 2005, HBoVs have garnered significant attention due to their widespread presence in clinical samples from respiratory and gastrointestinal tracts (14). The prevalence of HBoVs varies across reports from different countries, ranging from 1.3% to 63% in fecal samples and 1% to 56.8% in respiratory samples, with an estimated average global prevalence of 6% (15). Given their high infection prevalence worldwide, numerous studies have been conducted on HBoVs by researchers from various countries. However, most previously published studies have focused primarily on pediatric populations, with limited data covering a broader age range from infants to adults. This study represents a pioneering effort to collect substantial data from both pediatric and adult patients presenting with ILI. Since the discovery of HBoV, such a comprehensive investigation within both pediatric and adult populations in Kunming has not been undertaken, highlighting the significant contribution of our study to the understanding of HBoV-1 infection. Our findings support the role of HBoV-1 as an important pathogenic factor that warrants investigation in suspected ILI and respiratory tract infections, whether in isolation or conjunction with other pathogens. In

addition, bocavirus has emerged as a recently identified pathogen requiring thorough epidemiological analysis in the context of Kunming. Furthermore, there has been a lack of phylogenetic analyses to evaluate the distribution of different strains of the virus in this region. Therefore, the objectives of this study were to characterize HBoV-1 infection in children and adults over 1 month of age with ILI and to determine the phylogenetic characteristics and seasonality of HBoV-1 in Kunming.

## MATERIALS AND METHODS

### Patients and sample collection

Patients, including children and adults aged 1 month to 97 years, enrolled in this study sought medical attention for ILI from December 2017 to December 2023. ILI was defined according to the World Health Organization's (WHO) criteria, which includes a fever of ≥38°C accompanied by symptoms such as cough and/or sore throat. A total of 20,181 oropharyngeal swab (OPS) samples collected during the study were placed in the virus transport medium and promptly transferred to the Influenza Network Molecular Laboratory of the Yunnan Center for Disease Control and Prevention, where they were stored at −70°C until further analysis. The project was conducted as part of the Chinese surveillance system for ILI in two general hospitals: the First Affiliated Hospital of Kunming Medical University ($n = 7,208$) and Anning First People's Hospital ($n = 6,080$), along with a pediatric facility, the Children's Hospital ($n = 6,893$), in Kunming, China.

Demographic data were retrospectively analyzed, and data from all participants who tested positive for HBoV were included in subsequent analyses.

### Samples preparation and DNA extraction

The specimens were preserved at −70°C until the night before nucleic acid extraction when they were transferred to 4°C for thawing. Viral DNA and RNA were extracted from 200 µL of the preserved samples using an automatic nucleic acid extractor, following the manufacturer's instructions with TIANLONG Viral DNA/RNA Fast Extraction Kits (TIANLONG, Xi'an, China).

### Respiratory virus detection

Samples included in our study were tested for 18 respiratory viruses, including HBoV, using multiplex real-time fluorescence quantification PCR. Specific viral nucleic acids were detected for 17 viruses: influenza virus types A and B (Flu), respiratory syncytial virus (RSV), human rhinoviruses (HRV), adenoviruses (AdV), human metapneumovirus (HMPV), human coronaviruses NL63, 229E, HKU1, OC43, MERS, and SARS (HCoV), and parainfluenza virus types 1–4 (PIV), with SARS-CoV-2 detection added in 2022. All primers and probes for the viruses tested were obtained from the China CDC and were validated for sensitivity and accuracy using randomly selected samples from commercial kits prior to the study. The multiplex real-time fluorescence PCR mixture was prepared in a total volume of 20 µL, consisting of 5 µL of Fast 1-Step Mix (4×) (Applied Biosystems, USA), 5 µL of viral DNA/RNA, 0.4 µM forward primer, 0.4 µM reverse primer, and 0.2 µM probe for each pathogen, adjusted to a final volume of 20 µL with nuclease-free water. Real-time PCR was conducted at 50°C for 5 minutes and 95°C for 20 seconds, followed by 40 cycles of 95°C for 15 seconds and 55°C for 45 seconds on the Bio-Rad CFX96 Real-Time PCR System, with fluorescence detection occurring at 55°C.

### Next-generation sequencing

HBoV nucleic acid was re-extracted from 200 µL of all positive samples for NGS using the BioPerfectus Viral Nucleic Acid Isolation Kit (BioPerfectus, Jiangsu, China), according to the manufacturer's recommendations. The near-whole genome of HBoV was synthesized from three overlapping fragments using Platinum SuperFi II PCR Master Mix

(Invitrogen, USA) according to previously described methods (16). Independent PCRs were conducted at varying annealing temperatures, and the PCR products were mixed and sequenced after all reactions were completed for each sample. The sequences of the primers and their respective annealing temperatures are detailed in Table 1. The PCR mixtures consisted of 2 × Platinum SuperFi II PCR Master Mix, 0.5 µM of each primer, 5 µL of extracted DNA, and RNase-free water adjusted to a final volume of 50 µL. The PCR conditions were as follows: 98°C for 30 seconds, followed by 35 cycles of 98°C for 10 seconds, 61°C or 63°C for 30 seconds, and 72°C for 1 minute, with a final extension at 72°C for 7 minutes.

PCR amplicon mixtures were purified using the QIAGEN MinElute PCR Purification Kit (QIAGEN, Germany) and quantified with the Invitrogen Qubit dsDNA HS Assay Kit (ThermoFisher, USA). Libraries were prepared using the Illumina Nextera XT DNA Library Preparation Kit (Illumina, USA). Constructed libraries were pooled and subsequently sequenced on a MiSeq sequencer using the MiSeq Reagent Kit v2 (300 cycles, Illumina, USA).

## Sequence analysis

The quality of raw reads was assessed using CLC Genomics Workbench software v23 (CLC v23), and trimming was performed as necessary. Paired-end reads were mapped to the following reference sequences: HBoV-1 isolate st1 (acc. No. DQ000495), HBoV-2 isolate KU1 (acc. No. GQ200737), HBoV-3 strain W471 (acc. No. EU918736), and HBoV-4 isolate HBoV4/ETH_P3/2016 (acc. No. MG383446); consensus sequences were then exported for further analysis. Reads with lower coverage that mismatched the reference sequences were *de novo* assembled, and contigs were compared against the database using BLAST (https://blast.ncbi.nlm.nih.gov/Blast.cgi) to enhance alignments. All read analyses were conducted using CLC v23.

## Phylogenetic analysis

Representative HBoV-1 reference sequences from around the globe, with a focus on isolates from previous strains identified in China, were obtained from GenBank for constructing alignments and phylogenetic trees. Sequences belonging to genotypes HBoV-2, HBoV-3, and HBoV-4, as described in the literature, were included as outgroups in the analysis. Alignments were performed using MAFFT online software (version 7) (https://mafft.cbrc.jp/alignment/server/index.html). Aligned sequences were manually edited, and phylogenetic trees were generated using the Maximum Likelihood method based on the General Time Reversible model with a Gamma distribution and Invariant sites (G + I) pattern, both implemented in MEGA software (version 7). Bootstrap probabilities for 1,000 iterations were calculated to assess the reliability of individual nodes in each phylogenetic tree. The layout of all trees was performed using the online software TVBOT, and the final visualization was enhanced using TVBOT (version 2.6) (https://chiplot.online/tvbot.html).

TABLE 1 Primers' sequences and PCR annealing temperature

|  | Primer | Sequence(5′–3′) | Position[a] | Annealing(°C) |
|---|---|---|---|---|
| Pair1 | HBoV-1 F | GCCGGCAGACATATTGGATT | 1–20 | 63 |
|  | HBoV-1 R | GCCACCAACAACCGCGTAGAT | 1,789–1,809 |  |
| Pair2 | HBoV-2 F | TTACGGGCCTGCYTCAACAG | 1,515–1,534 | 61 |
|  | HBoV-2 R | CTGGATCCAATAATTCCACCAA | 3,282–3,303 |  |
| Pair 3 | HBoV-3 F | CATGGAAGCAGATGCCTCC | 3,045–3,063 | 63 |
|  | HBoV-3 R | CGGCTAGGTTCGAGACGG | 5,195–5,212 |  |

[a]By reference to GenBank sequence KP710213.

## Statistical analysis

The variables of gender and age were assessed using the $\chi^2$ test or Fisher's exact test, with a *P*-value of less than 0.05 considered as the threshold for statistical significance. Statistical analyses were performed using SPSS software version 26.0 (SPSS Inc.).

## RESULTS

### Prevalence and epidemiology of HBoV infection

Out of 20,181 specimens, 168 (0.8%) tested positive for HBoV. Among the 168 HBoV-positive participants, 165 (98.2%) were children under 18 years old, while 3 (1.8%) were adults. The prevalence was 1.0% (165/16,406) in children and 0.1% (3/3,775) in adults, with the detection rate in pediatric patients (<18 years old) significantly higher than that in adults (≥18 years old) (*P* < 0.001). According to age groups (Table 2), the positivity rates were 0.5% in children under 1 year old, 1.9% in children aged 1~ years, 1.6% in those aged 3~ years, 0.2% in children aged 5~ years, and 0.1% in those aged 18~ years. The differences in positivity rates among the various age groups were statistically significant ($\chi^2$=137.820, *P* < 0.001). A pairwise comparison with Bonferroni's adjustment indicated that the highest positivity rates were observed in the 1~ and 3~ year-old age groups, showing a linear correlation with decreasing positivity rates in older ages ($\chi^2$=35.225, *P* < 0.001). Among all HBoV-positive patients, 92.9% (156/168) were under 5 years old (Fig. 1).

Among HBoV-positive children, 81.2% (134/165) had HBoV as a single infection, while 18.8% (31/165) experienced co-infections with other respiratory viruses, with RSV being the most commonly observed co-infection (six children) (Fig. 2). No co-infections were identified in adults.

In the gender analysis (Table 2), HBoV was detected more frequently in males (*n* = 99/11,233; 0.9%) than in females (*n* = 69/8,948; 0.8%); however, this difference was not statistically significant (*P* = 0.392).

### Bocavirus case frequency and seasonality

Based on the sampling period (Fig. 3), the years 2018 and 2023 exhibited significantly higher numbers of bocavirus cases (Fig. 3A). The HBoV cases in 2023 were 2.7 times higher than those in 2020. The seasonal distribution of tested HBoVs is shown in Fig. 3B. Climate and weather did not significantly impact the frequency of bocavirus cases.

### Sequencing results

Out of the 168 HBoV-positive samples, 145 (86.3%) were available for further analysis, with 129 sequences allowing for type determination. Among the 145 samples subjected to PCR amplification, 107 produced amplicons that nearly covered the full HBoV genome, while 22 produced amplicons covering two-thirds of the genome (Fig. 4). As expected, all samples mapped to the HBoV-1 reference sequence (acc. No. DQ000495). Overall,

**TABLE 2**  Characterization of HBoV positivity and negativity in ILI cases by gender and age groups

| | Total | HBoV negative | HBoV positive | P value |
|---|---|---|---|---|
| | (*n* = 20181) | (*n* = 20013) | (*n* = 168) | |
| Gender | | | | 0.392 |
| Male | 11,233 | 11,134 (99.1%) | 99 (0.9%) | |
| Female | 8,948 | 8,879 (99.2%) | 69 (0.8%) | |
| Age group (years) | | | | < 0.001 |
| <1 | 3,744 | 3,726 (99.5%) | 18 (0.5%) | |
| 1~ | 4,385 | 4,303 (98.1%) | 82 (1.9%) | |
| 3~ | 3,496 | 3,440 (98.4%) | 56 (1.6%) | |
| 5~ | 4,781 | 4,772 (99.8%) | 9 (0.2%) | |
| 18~ | 3,775 | 3,772 (99.9%) | 3 (0.1%) | |

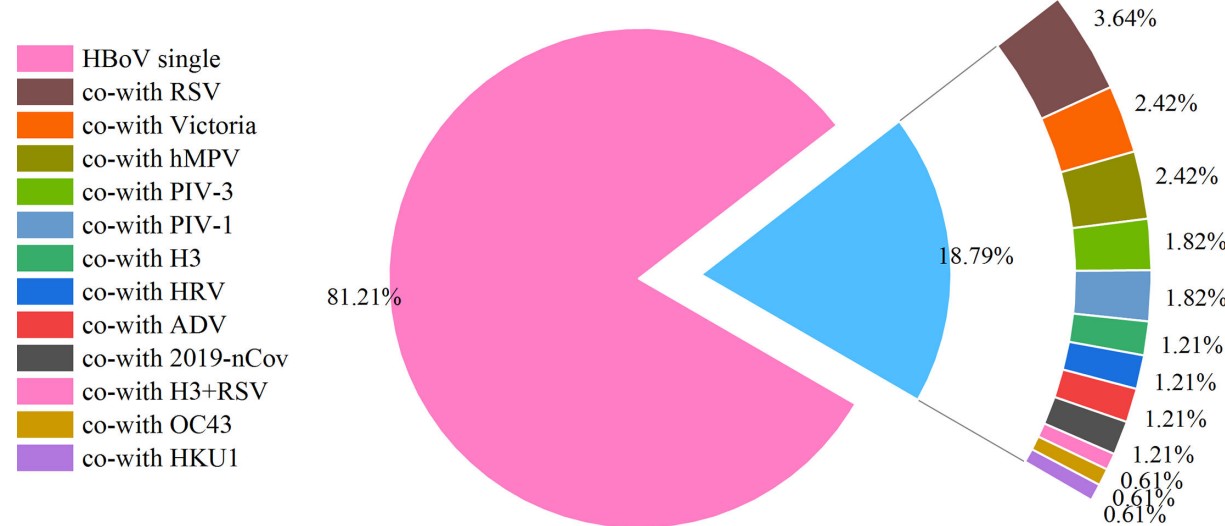

**FIG 1** Proportion of HBoV single infections and co-infections with other viruses in children.

most samples yielded consensus sequences covering the entire amplified region with a minimum coverage exceeding 100×. For samples with low coverage, *de novo* assembly was performed, and BLAST was utilized to search for contigs longer than 600 bp. In total, human sequences comprised the majority of the longer contigs, limiting the further improvement of the assembly.

## Phylogenetic and diversity analysis

We performed phylogenetic analyses using whole-genome sequences (Fig. 5A), as well as the NS1 (Fig. 5B), NP1 (Fig. 5C), and VP1 (Fig. 5D) regions. The VP region encodes the VP1, VP2, and VP3 proteins, with the VP1 gene being the longest; VP2 and VP3 completely overlap within the VP1 gene, but the initiator codons for the three genes are located at different positions. Therefore, the evolutionary tree based on the VP1 gene was used to represent the relationships among all three genes. The consensus whole-genome sequences of all HBoV-1 strains obtained in this study, excluding identical concordant sequences, have been deposited in GenBank under acc. Nos. PP625017-PP4625123.

The phylogenetic tree of the complete genome sequences demonstrated that all HBoV-1 sequences, whether detected singly or in co-detection with other viruses, clustered together, indicating that this lineage has dominated most of China in recent years, without the formation of additional branches or time-specific branches (Fig. 5).

We analyzed homology at both the nucleotide and amino acid levels for each coding region of the genome in comparison to the reference nucleotide sequence DQ000496. The nucleotide identities among our samples ranged from 98.98% to 99.96% when the coding regions were aligned, indicating that the VP coding region exhibited the greatest variability. The identities for the complete HBoV-1 genome were between 98.97% and 99.94%. The evolutionary tree constructed based on the VP1 genes closely resembled that based on the whole genome. Nucleotide identities were 99.58% to 100% for the NS1 gene, 99.71% to 100% for the NP1 gene, and 98.62% to 99.9% for the VP gene.

Most nucleotide variants in the complete genome were conserved at the amino acid level, resulting in a divergence of 0.06% to 1.85% among all Kunming isolates. The NP1 protein, having the fewest mutations—seven—was the most conserved, while NS1, the longest of the NS proteins, had eight variable sites. The VP region had the highest number of amino acid substitutions, totaling 20, consistent with the nucleotide variation (Table 3). The amino acid substitution site N474S in the VP1 protein (corresponding to N345S in the VP2 protein) was present in all 107 whole-genome sequences in our

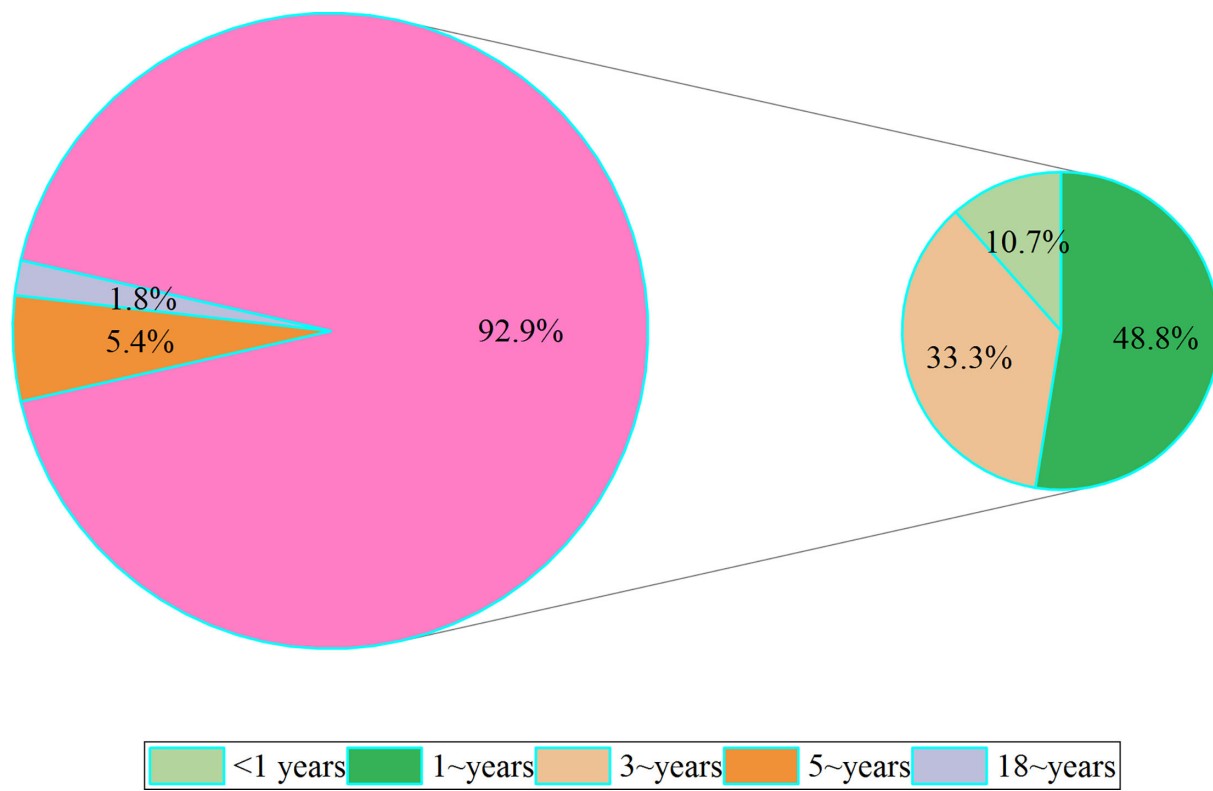

**FIG 2** Proportion of HBoV-positive cases in different age groups.

study. In addition, a total of 35 amino acid mutation sites were found in all CDS (coding sequence) regions of 52 sequences among the 107 sequences. Among the 52 sequences, in 50 sequences of them, there were only 1 to 3 amino acid substitutions in each sequence. Notably, two strains exhibited more than three amino acid substitution sites: nine in isolate KM HBoV-1–29 (2019) and six in KM HBoV-1–175 (2023), both sharing three amino acid mutation sites in the VP1 protein: L40S, G415S, and F540Y.

## DISCUSSION

Respiratory infections are commonly caused by various pathogens such as bacteria, viruses, and fungi. In the past decades, well-known respiratory viruses like influenza or adenovirus were often considered primary suspects when diagnosing respiratory illnesses. However, with advancements in medical research and technology, emerging viruses like HBoV-1 have been identified as potential culprits. HBoV-1 was a

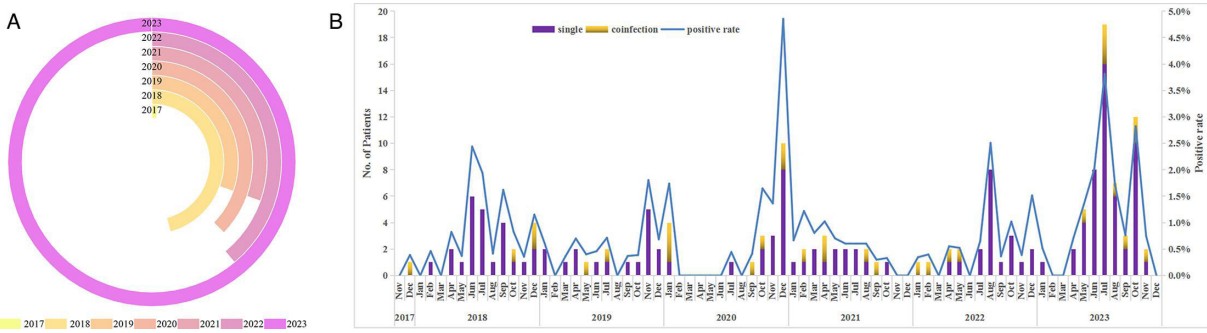

**FIG 3** (A) Distribution of HBoV-positive cases, 2017–2023. (B) Seasonal distribution of HBoV from December 2017 and December 2023. Temporal distribution of HBoV from December 2017 and December 2023.

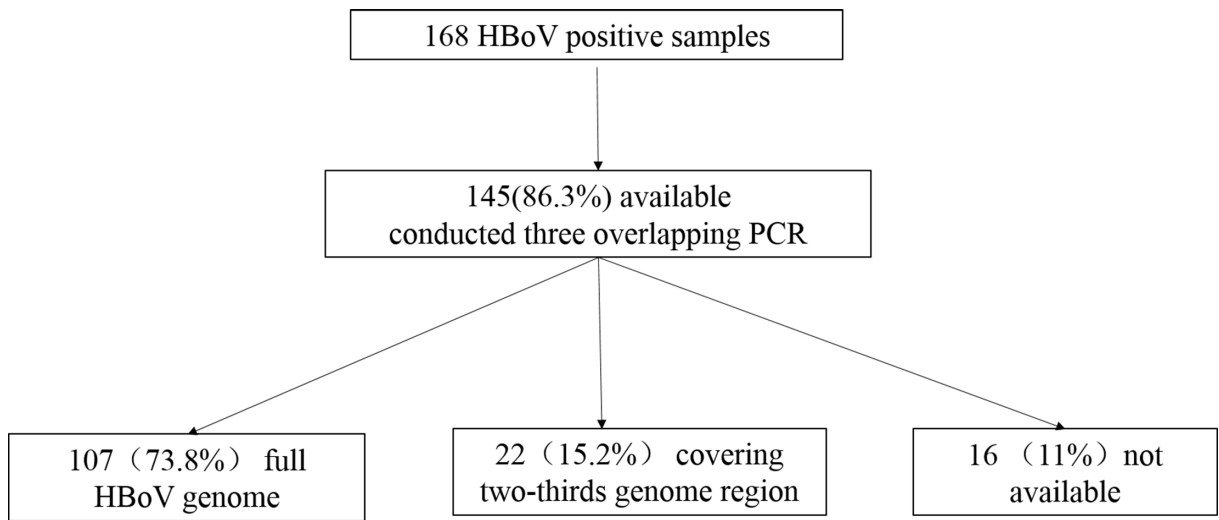

**FIG 4** Study flow chart.

correspondingly nascent virus that has only recently been identified and documented in the field of virology. Since its discovery, HBoV-1 has attracted the attention of scholars around the globe, and reports of respiratory and gastrointestinal diseases caused by it have emerged from all over the world. However, the majority of studies have focused primarily on infants and young children in outpatient, hospitalized ,or critically ill pediatric intensive care unit (PICU) (14, 16–19), with a subset of papers describing the characteristics of HBoV infections in adult patients (20–22), but only an extremely sporadic number of reports with regard to whole populations across a wide range of ages, including infants, young children, and adults (23, 24). Our study systematically analyzed the characteristics of HBoV-positive samples from a broadly age-distributed group of participants with ILI from Kunming, southwestern China for the first time. Consistent with previous findings (25, 26), detection rates were significantly higher in pediatric patients (<18 years of age) compared to adults (≥18 years of age), with the majority of HBoV-positive patients being <5 years of age and positivity rates decreasing with age. Numerous studies have demonstrated that the prevalence of HBoV-1 infection varies by age, but tends to be higher in very young children (27), especially those under 5 years (26). Therefore, it can be assumed that <5 years of age is a risk factor for HBoV infection. However, the probability of HBoV-1 infection decreased with age. The presumable reason for this fact was that seroepidemiologic data have suggested that by the age of 6 years, all children were likely to have been exposed to HBoV-1 (28), and therefore the corresponding protective antibodies might be present in the serum.

The prevalence of HBoV-1 among the 20181 ILI patients in Kunming was 0.83%. By contrast, a previously published cross-sectional nested study reported a higher detection rate of HBoV-1 in children and adults with ILI, yielding a positivity rate of 2.8% (23). The disparity in HBoV detection rates between studies may be attributed to regional variations in HBoV prevalence and the timing of the studies. Notably, nearly half of our study period coincided with the global COVID-19 pandemic, during which effective prevention and control measures in China (such as mask-wearing, hand hygiene, social distancing, lockdowns, and travel restrictions) likely contributed to a reduced HBoV positivity rate. In addition, variations in detection methods may play a role; nested PCR (29, 30) is generally more sensitive than real-time fluorescence PCR, potentially yielding detection rates exceeding 0.83%. Furthermore, the age and severity of the disease among study participants can significantly influence detection rates. Most studies reporting higher rates have focused on hospitalized or PICU children under 5 years old with acute severe respiratory illness (17, 18, 25, 31).

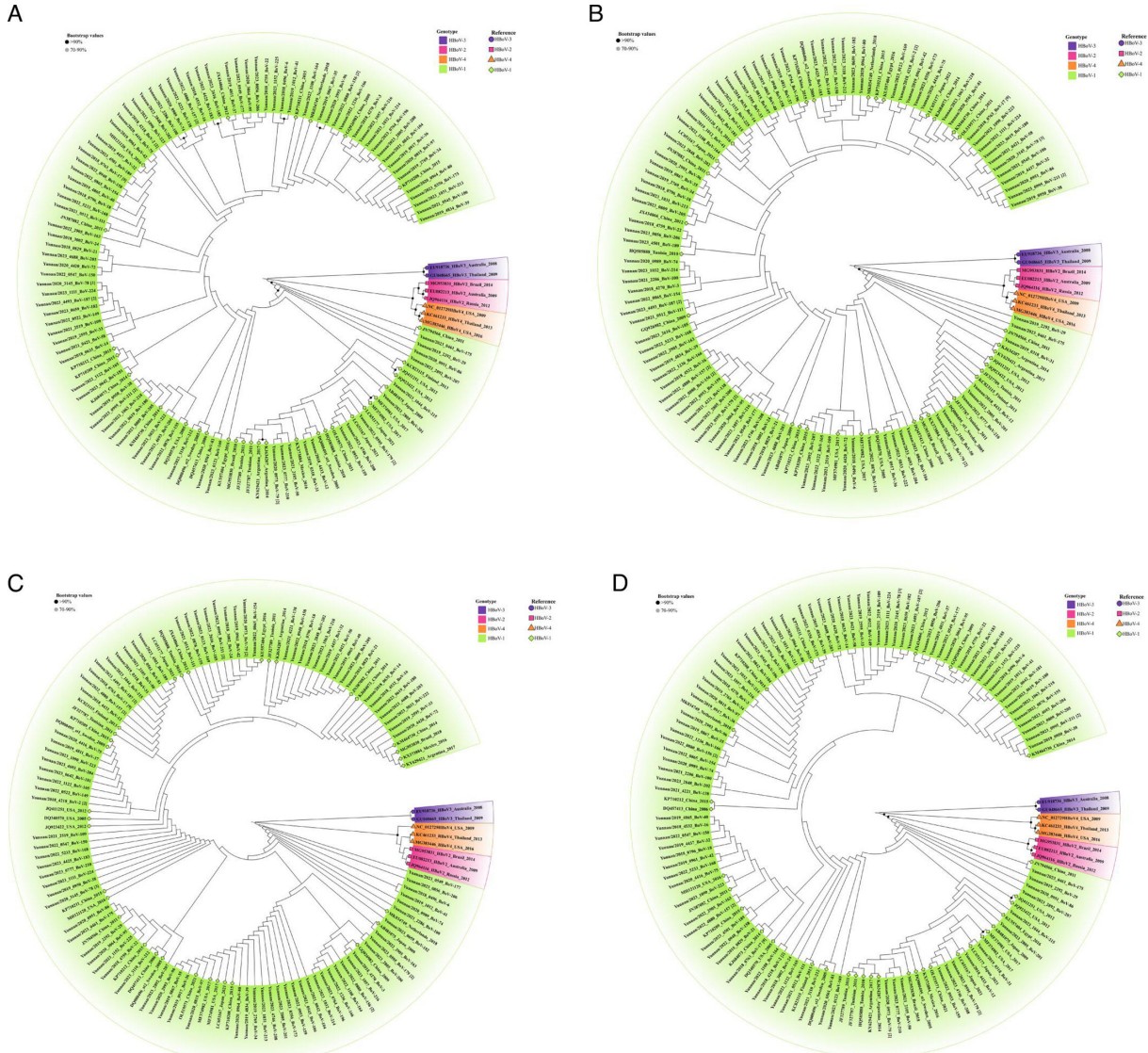

**FIG 5** (A) Phylogenetic tree of the whole genome. (B) Phylogenetic tree of the NS1 gene. (C) Phylogenetic tree of the NP1 gene. (D)Phylogenetic tree of the VP1 gene. Only sequences with unique nucleotide residues were used for phylogenetic analysis, with the numbers in brackets indicating the number of strains with identical sequences detected in our study. Phylogenetic analysis revealed that HBoV-1 was the most predominant epidemic strain in Kunming, China. The tree illustrates that the Kunming HBoV-1 sequences are closely related to strains isolated worldwide.

High co-detection rates are a well-recognized feature of bocavirus infection. HBoV-1 is one of the respiratory viruses usually recognized along with other such viruses (25, 26, 31). It is likely that at least one additional respiratory pathogen will be found in as many as 8.3% to 100% of HBoV-1 DNA-positive respiratory samples (15, 25). Our study was consistent with previous findings that HBoV-1 was co-detected with another respiratory virus in 18.8% of cases. The high co-infection rate of this virus may be due to its long-lasting persistence of shedding after initial infection (25, 26), which has been reported to persist in the respiratory mucosa for up to 6 months (32). Such persistent persistence and viral shedding increase the likelihood of co-infection with additional pathogens. As a result, considerable debate exists on establishing HBoV as a causative agent of acute respiratory infections following qualitative detection of viral DNA in nasopharyngeal swab (NPS) samples (33). A limitation of our study is that detection of HBoV-1 and co-infecting agents was conducted using real-time PCR, without pathogen isolation through culture, which restricts the ability to infer etiological causality.

**TABLE 3** Amino acid substitution sites of HBoV proteins

| Isolate | Substitution sites of amino acid | | | | | | | | | | | | | | | | | | | | | | | | | | | | | | | | | | |
|---|---|---|---|---|---|---|---|---|---|---|---|---|---|---|---|---|---|---|---|---|---|---|---|---|---|---|---|---|---|---|---|---|---|---|---|
|  | NP1 protein | | | | | | | NS1 protein | | | | | | | | VP1 protein | | | | | | | | | | | | | | | | | | | |
| DQ000496 | 36 | 47 | 53 | 59 | 68 | 79 | 139 | 37 | 120 | 382 | 400 | 432 | 441 | 580 | 633 | 17 | 40 | 51 | 64 | 68 | 72 | 114 | 132 | 149 | 272 | 335 | 415 | 425 | 503 | 540 | 546 | 560 | 590 | 631 | 648 |
|  | R | R | H | S | K | S | D | L | V | D | A | M | R | D | G | R | L | S | E | D | D | G | D | T | A | A | G | S | A | F | N | I | S | M | T |
| KM HBoV-1-3 2018 |  |  |  | N |  | N |  |  |  |  |  |  |  |  |  |  |  |  |  |  |  |  |  |  |  |  |  |  |  |  |  |  |  |  |  |
| KM HBoV-1-6 2018 |  |  |  | N |  | N |  |  |  |  |  |  |  |  |  |  |  |  |  |  |  |  |  |  |  |  |  |  |  |  |  |  |  |  |  |
| KM HBoV-1-12 2018 |  |  |  |  |  |  |  |  |  |  |  |  |  |  |  | K |  |  |  |  |  |  |  |  |  |  |  |  |  |  |  |  | T |  |  |
| KM HBoV-1-16 2018 |  |  |  |  |  |  |  |  |  |  |  |  |  |  |  |  |  |  |  |  |  |  | N |  |  |  |  |  |  |  |  |  |  |  |  |
| KM HBoV-1-22 2018 |  |  |  |  |  |  |  |  |  |  |  |  |  |  |  |  | S |  |  |  |  |  |  |  |  |  |  |  |  |  |  |  |  |  |  |
| KM HBoV-1-29 2019 | S |  | R | N | R | N |  |  |  |  |  |  |  |  |  |  | S |  |  |  |  |  |  |  |  |  | S |  |  |  |  |  |  |  |  |
| KM HBoV-1-31 2019 |  |  |  |  |  |  |  |  |  |  |  |  |  |  |  | K |  |  |  | N |  |  |  |  |  |  |  |  |  | Y | H |  | T |  |  |
| KM HBoV-1-32 2019 |  |  |  |  |  |  |  |  |  |  |  |  |  |  |  |  |  |  |  |  |  | S |  |  |  |  |  |  |  |  |  |  |  |  |  |
| KM HBoV-1-35 2019 |  |  |  |  |  |  |  |  |  |  |  | I |  |  |  |  |  |  |  |  |  |  |  |  |  |  |  |  |  |  |  |  |  |  |  |
| KM HBoV-1-37 2019 |  |  |  |  |  |  |  |  |  |  |  |  |  |  |  |  | S |  |  |  |  |  |  |  |  |  |  |  |  |  |  |  |  |  |  |
| KM HBoV-1-38 2019 |  |  |  |  |  |  |  | I |  |  |  |  |  |  |  |  |  |  |  |  |  |  |  |  |  |  |  |  |  |  |  |  |  |  |  |
| KM HBoV-1-39 2019 |  |  |  |  |  |  |  |  |  |  |  |  |  |  |  |  |  |  |  |  |  |  |  |  |  |  | N |  |  |  |  |  |  |  |  |
| KM HBoV-1-41 2019 |  |  |  |  |  | N |  |  |  |  |  |  |  |  |  |  |  |  |  |  |  |  |  |  |  |  |  |  |  |  |  |  |  |  |  |
| KM HBoV-1-42 2019 |  |  |  |  |  |  |  |  |  |  |  |  |  |  |  |  |  |  | K |  |  |  |  |  |  |  |  |  |  |  |  |  |  |  |  |
| KM HBoV-1-50 2021 |  |  |  |  |  |  |  |  |  |  |  |  |  |  |  |  |  |  |  |  |  |  |  |  |  |  |  |  |  |  |  |  | T |  |  |
| KM HBoV-1-74 2020 |  |  |  |  |  | N |  |  |  | N |  |  |  |  |  |  |  |  |  |  |  |  |  |  |  |  |  |  |  |  |  |  |  |  |  |
| KM HBoV-1-79 2020 |  |  |  |  |  |  |  |  |  |  |  |  |  |  | R |  |  |  |  |  |  |  |  |  |  |  |  |  |  |  |  |  | T |  |  |
| KM HBoV-1-86 2020 |  |  |  |  |  |  |  |  |  |  |  |  |  |  |  |  | S |  |  |  |  |  |  |  |  |  |  |  |  |  | H |  |  |  |  |
| KM HBoV-1-95 2020 |  |  |  |  |  |  |  |  |  |  |  |  |  |  |  |  | S |  |  |  |  |  |  |  |  |  |  |  |  |  |  |  |  |  |  |
| KM HBoV-1-97 2020 |  |  |  |  |  |  |  |  |  | N |  |  |  |  |  |  |  |  |  | N |  |  |  |  |  |  |  |  |  |  |  |  |  |  |  |
| KM HBoV-1-100 2021 |  |  |  |  |  | N |  |  |  |  |  |  |  |  |  |  |  |  |  |  |  |  |  |  |  |  |  |  |  |  |  |  |  |  |  |
| KM HBoV-1-109 2021 |  |  |  |  |  |  |  |  |  |  |  |  |  | N |  |  |  |  |  |  |  |  |  |  |  | T |  |  |  |  |  |  |  |  |  |
| KM HBoV-1-138 2021 |  |  |  |  |  |  |  |  |  |  |  |  | K |  |  |  |  |  |  |  |  |  |  |  |  |  |  |  |  |  |  |  |  |  |  |
| KM HBoV-1-150 2022 |  |  |  |  |  |  |  |  |  |  |  |  |  |  |  |  |  |  |  |  |  |  |  |  |  |  |  |  |  |  |  |  |  |  |  |
| KM HBoV-1-154 2022 | K |  |  |  |  |  |  |  |  |  |  |  |  |  |  |  |  |  |  |  |  |  |  |  |  |  |  |  |  |  |  |  |  |  |  |
| KM HBoV-1-156 2022 |  |  |  |  |  | N |  |  |  |  |  |  |  |  |  |  |  |  |  |  |  |  |  |  |  |  |  |  |  |  |  |  |  |  |  |
| KM HBoV-1-159 2022 |  |  |  |  |  |  |  |  |  |  |  |  |  |  |  | K |  |  |  |  |  |  |  |  |  |  |  |  |  |  |  |  | T |  |  |
| KM HBoV-1-163 2022 |  | K |  |  |  |  |  |  |  |  |  |  |  |  |  |  |  |  |  |  |  |  |  |  |  |  |  |  |  |  |  |  |  |  |  |
| KM HBoV-1-164 2022 |  |  |  |  |  | N |  |  |  |  |  |  |  |  |  |  |  |  |  |  |  |  |  |  |  |  |  |  |  |  |  |  |  |  |  |
| KM HBoV-1-166 2022 |  |  |  |  |  | N |  |  |  |  |  |  |  |  |  |  |  |  |  |  |  |  |  |  |  |  |  |  |  |  |  |  |  |  |  |
| KM HBoV-1-172 2022 |  |  |  |  |  |  |  |  |  |  |  |  |  |  |  |  |  |  |  |  |  |  |  |  |  |  |  |  |  |  |  |  | T |  | I |
| KM HBoV-1-174 2023 |  |  |  |  |  | N |  |  |  |  |  |  |  |  |  |  |  |  |  |  |  |  |  |  |  |  |  |  |  |  |  |  |  |  |  |
| KM HBoV-1-175 2023 |  |  |  |  |  |  |  |  |  |  |  |  |  |  |  |  | S |  |  |  |  |  |  | N |  |  | S |  |  | Y |  | V |  | L |  |
| KM HBoV-1-177 2023 |  | K |  |  |  | N |  |  |  |  |  |  |  |  |  |  | S |  |  |  |  |  |  |  |  |  |  |  |  |  |  |  |  |  |  |
| KM HBoV-1-179 2023 |  |  |  |  |  |  |  |  |  |  |  |  |  |  |  | K |  |  |  |  |  |  |  |  |  |  |  |  |  |  |  |  | T |  |  |

**TABLE 3** Amino acid substitution sites of HBoV proteins (*Continued*)

| Isolate | NP1 protein | | | | | | | NS1 protein | | | | | | | | VP1 protein | | | | | | | | | | | | | | | | | | | | |
|---|---|---|---|---|---|---|---|---|---|---|---|---|---|---|---|---|---|---|---|---|---|---|---|---|---|---|---|---|---|---|---|---|---|---|---|
| | 36 | 47 | 53 | 59 | 68 | 79 | 139 | 37 | 120 | 382 | 400 | 432 | 441 | 580 | 633 | 17 | 40 | 51 | 64 | 68 | 72 | 114 | 132 | 149 | 272 | 335 | 415 | 425 | 503 | 540 | 546 | 560 | 590 | 631 | 648 |
| DQ000496 | R | R | H | S | K | S | D | L | V | D | A | M | R | D | G | R | L | S | E | D | D | G | D | T | A | A | G | S | A | F | N | I | S | M | T |
| KM HBoV-1-182 2023 | | | | | | | N | | | | | | | | | | | | | | | | | | | | | | | | | | | | |
| KM HBoV-1-185 2023 | | | | | | | | | | | | | | | | | | N | | | | | | | | | | | | | | | | | |
| KM HBoV-1-189 2023 | | | | | | | | | | | | | | | | K | | | | | | | | | | | | | | | | | | | |
| KM HBoV-1-196 2023 | | | | | | N | | | | | | | | | | | | | | | | | | | | | | | | | | | | | |
| KM HBoV-1-200 2023 | | | | | | N | | | | | | | | | | | | | | | | | | | | | | | | | | | | | |
| KM HBoV-1-201 2023 | | | | | | | | | | | | | | | | | | | | | | | | | | | | | | | | | T | | |
| KM HBoV-1-206 2023 | | | | | | N | | | | | | | | | | | S | | | | | | | | | | | | | | | | | | |
| KM HBoV-1-207 2023 | | | | N | | | | | | | | | | | | | | | | | | | | | | | | | | | H | | | | |
| KM HBoV-1-208 2023 | | | | | | | | | | | | | | N | | K | | | | | | | | | | | | | | | | | T | | |
| KM HBoV-1-210 2023 | | | | | | | | | | | | | | | | | | | | | | | | | | | | | | | | | T | | |
| KM HBoV-1-212 2023 | | | | | | | | | | | | | K | | | | | | | | N | | | | | | | | T | | | | | | |
| KM HBoV-1-213 2023 | | | | | | | | | I | | | | | | | | | | | | | | | | | | | | | | | | | | |
| KM HBoV-1-214 2023 | | | | | | N | | | | N | | | | | | K | | | | | | | | | | | | | | | | | | | |
| KM HBoV-1-215 2023 | | | | | | | | | | | | | | | | | | | | | | | | | T | | | | | | | | | | |
| KM HBoV-1-216 2023 | | | | N | | N | | | | | | | | | | | | | | | | | | | | | | | | | | | | | |
| KM HBoV-1-221 2023 | | | | | | | | | | | | | | | | K | | | | | | | | | | | | | | | | | | | |
| KM HBoV-1-225 2023 | | | | | | | | | | | T | | | | | | | | | | | | | | | | | | | | | | T | | |

Therefore, numerous studies have advocated for the use of quantitative PCR, serological, or mRNA assays as more reliable methods for the timely and accurate diagnosis of HBoV-1 infection in patients with ILI (27). However, first, threshold criteria for quantitative testing are difficult to determine; second, serologic testing may have limited sensitivity during episodes of viral infection because serologic conversion may not occur until late in the course of the disease during acute illness (3, 21); and lastly, mRNAs that are hallmarks of acute viral genome activity are consistently detectable even during periods of persistent HBoV-1 infection, while other viruses can also be detected in nearly 60% of patients (31). Therefore, there is an urgent need to develop a more sensitive, as well as timely and accurate, clinical assay for HBoV-1, in combination with the patient's clinical presentation, to determine its pathogenic role and to clarify whether it is a disease suspect, an exacerbation factor, or a bystander?

We investigated gender-associated differences in our study, but the detection rates between males and females did not show statistical significance. Gender differences in ILI have not been fully explored. In-depth studies may be essential to dissect the impact of male and female gender differences on HBoV infection.

Most studies reported that HBoV-1 is predominantly detected in winter and spring (15, 16, 18). However, our findings suggest that climate and weather did not significantly influence bocavirus case frequency, and circulating seasons varied from year to year. This aligns with observations from Gamiño-Arroyo et al. (23), who reported HBoV's presence year-round in Mexico City, with no significant relationship to temperature and humidity. By contrast, Chen et al. (34) noted a correlation between HBoV infections and climatic factors in China. Due to the relatively low positivity rates in our study, the seasonal trends of HBoV circulation in southwestern China remain unclear. A prospective multi-centric study involving a larger number of acute respiratory samples from high-incidence populations is essential to better understand the seasonal patterns of HBoV and its potential associations with meteorological factors in the region.

Few molecular epidemiological studies of HBoV have been carried out previously in Kunming, and there is a lack of complete viral sequences to provide temporal or spatial characterization of the virus. The molecular characterization of the HBoV-1 strain in Kunming could contribute to understanding its evolution. Our sequencing and phylogenetic analysis revealed that all samples collected during the period of this study belonged to HBoV-1, which was consistent with the findings of other studies (17, 18, 23, 35, 36) that suggest HBoV-1 is predominantly present in the respiratory tract. DNA from HBoV-1 has almost exclusively been found in samples taken from the respiratory tract, indicating that this virus is undoubtedly spread *via* the respiratory system (3, 25, 32), although their pathogenic role in respiratory diseases is unproven (16, 37).

Genome phylogenetic analysis indicated that all HBoV-1 strains in our study clustered closely with strains from various countries, including China, the United States, Japan, Brazil, Egypt, and the Netherlands. This suggests minimal genetic variability in the HBoV-1 genome, consistent with findings from previous studies (16, 23, 38, 39). Notably, the evolutionary tree based on the VP1 gene mirrored that of the complete genome, indicating that phylogenetic analysis of the VP region serves as a reliable representation of the overall genomic relationships.

The genes encoding VP proteins exhibited the greatest variability compared to the nonstructural proteins NS1 and NP1, aligning with previous studies. The significant sequence heterogeneity between HBoV-1 and the other three genotypes underscores the scientific validity of using the VP region for strain classification, reinforcing its role in distinguishing HBoV-1 from other variants (18).

Genetic mutation and recombination are major drivers of viral evolution (40, 41). The fact that HBoV-1 was highly conserved compared to the original reference strain DQ000496 may also explain why HBoV-1 was less prevalent in adults, possibly because immunity acquired by infection at a young age may be maintained throughout life. The low number of nucleotide mutation sites we detected in all isolates during the 6-year study period suggests that locally circulating HBoV-1 isolates had slight genetic variation

over a short period of time. However, many studies have shown that intragenotypic recombination seems to play an important role in the evolution of Bocaviruses (41), and whether or not recombination mutations occurred in the strains in the present study needs to be further investigated. Whether the amino acid mutations elaborated in this study cause changes in protein conformation needs to be further investigated.

The absence of systematic *in vitro* cellular and animal culture models presents a significant limitation in assessing whether the identified substitutions and variable sites in the tested strains affect replication, infectivity, virulence, or the reactivity of neutralizing anti-HBoV-1 antibodies. This is particularly relevant for the two strains with the highest levels of mutation in this study. Future research utilizing these models would be essential for elucidating the functional implications of these genetic variations.

## Conclusions

In conclusion, this study reinforces previous findings that HBoV-1 is the primary human bocavirus linked to respiratory infections. The observed prevalence emphasizes the need for ongoing surveillance and research into this virus. Future studies should aim to deepen our understanding of HBoV's epidemiology, clinical manifestations, and potential treatments. In addition, developing effective diagnostic tools and vaccines is crucial for preventing and controlling HBoV infections. Overall, these results enhance our understanding of HBoV and its implications for human health.

## ACKNOWLEDGMENTS

The authors declare that no funds, grants, or other support were received during the preparation of this manuscript.

Material preparation, conduct of experiments, and data analysis were performed by Y.S. and J.Z. Data collection and sample preparation was done by J.Z. L.J. and Y.C. assisted in completing the statistical data analysis. The first draft of the manuscript was written by Y.S. and all authors commented on previous versions of the manuscript. All authors read and approved the final manuscript.

## AUTHOR AFFILIATION

[1]Yunnan Center for Disease Control and Prevention, Kunming, Yunnan, China

## AUTHOR ORCIDs

Yanhong Sun http://orcid.org/0009-0001-4355-3479
Jienan Zhou http://orcid.org/0000-0002-1517-5044

## AUTHOR CONTRIBUTIONS

Yanhong Sun, Conceptualization, Formal analysis, Investigation, Visualization, Writing – original draft, Writing – review and editing | Lili Jiang, Formal analysis | Yaoyao Chen, Formal analysis | Zhaosheng Liu, Validation | Meiling Zhang, Validation | Xiaonan Zhao, Project administration | Xiaoyu Han, Data curation | Lifen Zhang, Investigation | Xiaoqing Fu, Project administration, Supervision | Jienan Zhou, Conceptualization, Data curation, Funding acquisition, Resources, Supervision, Validation, Writing – review and editing

## DATA AVAILABILITY

All demographic data for this study were sourced from the Chinese surveillance system for ILI, collected between December 2017 and December 2023. The datasetsdata sets analyzed during the current study are available from the corresponding author upon reasonable request. The corresponding author's email is: 1191087570@qq.com. It should be noted that some of the data may be subject to restrictions due to patient privacy concerns. However, we are committed to providing access to the data within the

boundaries of these limitations and in accordance with applicable laws and ethical guidelines.

In addition, all sequencing data were obtained from the Influenza Network Molecular Laboratory of the Yunnan Center for Disease Control and Prevention. The HBoV-1 whole-genome sequences obtained in this study have been deposited in GenBank (https://www.ncbi.nlm.nih.gov/genbank/) under accession numbers PP625017-PP625123. The accession numbers corresponding to each strain can be found in Supplementary data.

Please contact the author for any further inquiries regarding data availability.

## ETHICS APPROVAL

This project was conducted using residual clinical samples. Written informed consent was not required as patients participating in this study were completely anonymous and verbal informed consent was obtained from parents or guardians.

This article does not contain any research conducted by the authors on human participants or animals.

## ADDITIONAL FILES

The following material is available online.

### Supplemental Material

**Supplemental material (Spectrum01564-24-s0001.txt).** Supplemental data.

### Open Peer Review

**PEER REVIEW HISTORY (review-history.pdf).** An accounting of the reviewer comments and feedback.

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
