## [Reviewer comments · Microbiology Spectrum]

Microbiology Spectrum

Prevalence and Molecular Characterization of Human bocavirus-1 (HBoV-1) in Children and Adults with Influenza-like illness (ILI) from Kunming, Southwest China

YanHong Sun, Lili Jiang, yao chen, zhaosheng liu, Mei-Ling Zhang, xiao zhao, Xiaoyu Han, Lifan Zhang, Xiaoqing Fu, and Jienan Zhou

Corresponding Author(s): Jienan Zhou, Yunnan Center for Disease Control and Prevention

Review Timeline:

Submission Date:	June 27, 2024
Editorial Decision:	August 29, 2024
Revision Received:	October 10, 2024
Accepted:	October 22, 2024

Editor: Jie Wang

Reviewer(s): Disclosure of reviewer identity is with reference to reviewer comments included in decision letter(s). The following individuals involved in review of your submission have agreed to reveal their identity: Ali Hafedh Abbas (Reviewer #1); Nikhil Chakravarty (Reviewer #3)

Transaction Report:

DOI: <https://doi.org/10.1128/spectrum.01564-24>

Re: Spectrum01564-24 (Prevalence and Molecular Characterization of Human bocavirus-1 (HBoV-1) in Children and Adults with Influenza-like illness (ILI) from Kunming, Southwest China)

Dear Ms. Jienan Zhou:

Thank you for the privilege of reviewing your work. Below you will find my comments, instructions from the Spectrum editorial office, and the reviewer comments.

Revision Guidelines

Sincerely,
Jie Wang
Editor
Microbiology Spectrum

Reviewer #2 (Comments for the Author):

Introduction and Importance of the study e.g. line no. 45 authors have targeted one province of China, hence better to describe about global status of ILI, then China and then USA and/or USA or other country. English editing is necessary.

Reviewer #3 (Comments for the Author):

This manuscript details phylogenetic analysis and assessment of positivity rates of human bocavirus-1 (HBoV-1) among children and adults in three health centers in Kunming, China. This analysis of over 20,000 samples found an HBoV-1 positivity rate of 0.8% and that, of HBoV patients, most were positive for HBoV-1.

This manuscript is important and details a current gap in knowledge regarding this virus. That said, there are some comments that must be addressed:

Major Questions:

1. For those above 18 years of age and positive for HBoV, what was the age breakdown? Were they more geriatric or younger? This is an important consideration for further investigation.

2. Considering the HBoV-1 is largely found in fecal samples, why were fecal samples not assessed in this study alongside OPS samples? Have comparative studies been conducted to see whether OPS samples accurately capture HBoV infection as compared to fecal samples?

3. Considering differences in methodology between other studies, is there the possibility that you are either over- or under-capturing cases? What steps have you/can you take to either normalize or effectively compare your results to others?

4. Is the population assessed representative of that region of China or China as a whole? How can these data be interpreted for a larger population?

General Concerns:

1. Extensive grammatical editing is needed throughout the manuscript.

Response to Reviewers

Dear reviewers:

We sincerely thank you for carefully reviewing our paper and putting forward valuable comments and suggestions. We attach great importance to your opinions and hereby make the following responses:

Firstly, we are extremely grateful for your careful reading and review of the paper. The issues you mentioned are indeed those that we did not fully consider during the writing process, and we deeply apologize. In response to the specific issues you raised, we have carried out careful modifications and improvements. The following is our specific response to each issue:

Reviewer #2 (Comments for the Author):

Introduction and Importance of the study e.g. line no. 45 authors have targeted one province of China, hence better to describe about global status of ILI, then China and then USA and/or USA or other country. English editing is necessary.

Response to Reviewers#2:

Regarding the point you pointed out, we indeed have a problem of unclear logical expression. Therefore, after referring to the relevant literature that has been published before, we have made the following

modifications as required: It is estimated that there are as many as several billion cases of influenza-like illness globally every year. According to the monitoring data in China in 2023, there are approximately 17 million cases of influenza-like illness across the country. In the United States, 9-49 million people are affected by ILI every year.

Reviewer #3 (Comments for the Author):

Response to Reviewers#3:

Major Questions:

1. For those above 18 years of age and positive for HBoV, what was the age breakdown? Were they more geriatric or younger? This is an important consideration for further investigation.

Response of question1:

Regarding the question you raised, we have conducted a review of relevant literature and found that for those cases testing positive for HBoV aged 18 and above, the age of exposure tends to cluster between 18~60 years old. In patients over 60 years old, HBoV is rarely or almost undetectable in respiratory or fecal samples. Additionally, our study included three adult cases with ages of 27, 35, and 50 respectively, which aligns with other research findings. This finding indicates a direction for future research design; whether individuals over the age of 60 should be excluded from studies on HBoV infection requires careful consideration

in research design.

2. Considering the HBoV-1 is largely found in fecal samples, why were fecal samples not assessed in this study alongside OPS samples? Have comparative studies been conducted to see whether OPS samples accurately capture HBoV infection as compared to fecal samples?

Response of question2:

In response to your question, our team members strongly agree and believe that it is an important approach for future research on HBoV. By comparing the positive detection rates of OPS and fecal samples for the same case, we can further determine the optimal sampling method for HBoV infection. We will actively consider your valuable suggestion in our upcoming research on HBoV.

The reason why this study did not compare the capture rates of OPS and fecal samples is because the initial design of this study was based on the following considerations: as our laboratory mainly focuses on influenza monitoring work, only OPS samples are involved in the monitoring, without including fecal samples. In routine monitoring work, we often find that a large proportion of ILI cases sent by sentinel hospitals test negative for influenza viruses. Therefore, we often consider what other pathogens may cause ILI cases and their incidence rate, age distribution, etc. Thus, for samples sent by sentinel hospitals, while

conducting influenza virus testing, we also conducted testing for 17 other viruses including HBoV to understand their activity in Kunming area and provide data support for formulating prevention and control strategies. As for collecting fecal samples, we indeed did not fully consider it at the beginning of the study due to three main reasons: patient cooperation; difficulty in collecting fecal samples compared to OPS; economic cost issues due to long duration and large sample size involved in this experiment. Therefore, fecal sample research was not included in the current study.

3. Considering differences in methodology between other studies, is there the possibility that you are either over- or under-capturing cases? What steps have you/can you take to either normalize or effectively compare your results to others?

Response of question3:

After reviewing the literature, our research findings indicate that there are indeed certain differences in the detection rate of HBoV compared to other studies, both in children and adult ILI cases. However, the primers and probes used in our study have been validated through parallel experiments using two or more commercial reagent kits and ThermoFisher microfluidic chips prior to conducting formal experiments, ensuring the reliability of our results.

As for the discrepancies observed compared to other research findings, it is true that our study has several limitations as you mentioned. We have analyzed these limitations in the discussion section of our paper. In future studies, we will strive for a more comprehensive design and planning from research design to inclusion criteria for samples, data collection methods, and experimental techniques if there are any other inadequately considered issues.

4. Is the population assessed representative of that region of China or China as a whole? How can these data be interpreted for a larger population?

Response of question4:

Regarding your question about whether the population assessed in our study represents the specific region or entire China, the population included in our research only represents the prevalence of HBoV among individuals in Kunming. It cannot represent China as a whole. This is because, firstly, we have reviewed previously published literature within China and found that the prevalence of HBoV varies across different regions, including capture rates and seasonality. Therefore, inferring nationwide or global prevalence based on one specific region lacks scientific validity for studying HBoV. Secondly, the three sentinel hospitals included in this study are large comprehensive hospitals with

strong capabilities in Yunnan Province. Based on collected case data, patients from all over Kunming City were treated at these hospitals; thus they can represent the prevalence of HBoV in this area.

As for how to interpret these data for a larger population? We believe that our research results are consistent with those already published in several aspects: Firstly, HBoV primarily infects children and has a higher incidence rate among those under 5 years old compared to adults. Secondly, there is a high occurrence rate of co-infection between HBoV and other viruses. Thirdly, HBoV is a DNA virus with relatively conserved genomes and low mutation rates.

Response to Reviewers#2 and #3:

The English expressions and grammar of the entire manuscript have been edited and revised.

During the revision process, we have also further optimized the structure of the paper to enhance the logic and readability of the article. We sincerely hope that these modifications and improvements can meet your requirements and make the paper more complete and persuasive.

Once again, thank you very much for your review comments and suggestions. We sincerely hope that you will be satisfied with the modifications and improvements we have made to the paper. If you have

other comments or suggestions, we are also very willing to listen. Thank you for your help and support.

Thank you and best regards.

Yours sincerely,

Yanhong Sun

Yunnan Center for Disease Control and Prevention, Kunming, Yunnan,
China

Corresponding author:

Name: Jienan Zhou

E-mail:1191087570@qq.com

Date: September 20, 2024

Re: Spectrum01564-24R1 (Prevalence and Molecular Characterization of Human bocavirus-1 (HBoV-1) in Children and Adults with Influenza-like illness (ILI) from Kunming, Southwest China)

Dear Ms. Jienan Zhou:

Your manuscript has been accepted, and I am forwarding it to the ASM production staff for publication. Your paper will first be checked to make sure all elements meet the technical requirements. ASM staff will contact you if anything needs to be revised before copyediting and production can begin. Otherwise, you will be notified when your proofs are ready to be viewed.

Sincerely,
Jie Wang
Editor
Microbiology Spectrum